# Molecular and Potential Regulatory Mechanisms of Melanin Synthesis in *Harmonia axyridis*

**DOI:** 10.3390/ijms21062088

**Published:** 2020-03-18

**Authors:** Da Xiao, Xu Chen, Renbin Tian, Mengmeng Wu, Fan Zhang, Liansheng Zang, James D. Harwood, Su Wang

**Affiliations:** 1Institute of Plant and Environment Protection, Beijing Academy of Agricultural and Forestry Sciences, Beijing 100097, China; xiaoda@ipepbaafs.cn (D.X.); 18754880390@163.com (X.C.); Wumeng1918@163.com (M.W.); zf6131@263.Net (F.Z.); jd_harwood@hotmail.com (J.D.H.); 2Jilin Engineering Research Center of Resource Insects Industrialization/Institute of Biological Control, Jilin Agricultural University, Changchun 130118, China; tianrb96@163.com (R.T.); lsz0415@163.com (L.Z.)

**Keywords:** tyrosine hydroxylase, DOPA decarboxylase, dopamine melanin, *Pannier*, *Harmonia axyridis*

## Abstract

Melanization is a common phenomenon in insects, and melanin synthesis is a conserved physiological process that occurs in epidermal cells. Moreover, a comprehensive understanding of the mechanisms of melanin synthesis influencing insect pigmentation are well-suited for investigating phenotype variation. The Asian multi-colored (Harlequin) ladybird beetle, *Harmonia axyridis*, exhibits intraspecific polymorphism based on relative levels of melanization. However, the specific characteristics of melanin synthesis in *H. axyridis* remains elusive. In this study, we performed gene-silencing analysis of the pivotal inverting enzyme, tyrosine hydroxylase (TH), and DOPA decarboxylase (DDC) in the tyrosine metabolism pathway to investigate the molecular and regulatory mechanism of melanin synthesis in *H. axyridis*. Using RNAi of *TH* and *DDC* genes in fourth instar larvae, we demonstrated that dopamine melanin was the primary contributor to the overall body melanization of *H. axyridis*. Furthermore, our study provides the first conclusive evidence that dopamine serves as a melanin precursor for synthesis in the early pupal stage. According to transcription factor *Pannier*, which is essential for the formation of melanic color on the elytra in *H. axyridis*, we further demonstrated that suppression of *HaPnr* can significantly decrease expression levels of *HaTH* and *HaDDC*. These results in their entirety lead to the conclusion that transcription factor *Pannier* can regulate dopamine melanin synthesis in the dorsal elytral epidermis of *H. axyridis*.

## 1. Introduction

The Asian multi-colored (Harlequin) ladybird beetle, *Harmonia axyridis* (Coleoptera: Coccinellidae), has been widely used for biological control of pests in fields and greenhouse crops for a long time [1]. It also exhibits intraspecific polymorphism based on the influence of melanization with over 200 different elytral color forms [2]. This striking intraspecific variation has prompted investigation into its genetic, biochemical, and evolutionary meaning [3,4] and forms the basis of the current study. In recent years, an association between melanization of *H. axyridis* and different phenotypes have been investigated, including variation in fitness parameters [5], prey capacity [6,7], behavioral characteristics [8], aggregation behavior [9], assortative mating [10], fertility [11], and responses to insecticide stress [12]. In addition, the degree of melanization has a linear relationship with temperature variation in *H. axyridis* [13]; for instance, melanization increased linearly with lower temperatures [14] and decreased linearly with elevated temperature [15]. The abovementioned research also highlighted that different phenotypic individuals exhibited varying behavioral traits. Thus, elucidating the molecular mechanism of melanin synthesis in *H. axyridis* would provide crucial information on the sole ontogenies and evolutionary histories of the attendant melanization.

Melanin is the final product in the melanization process and has a prominent role in wound healing, cuticle sclerotization, innate immunity, defensive reactions, cuticular coloration, and camouflage in insects [16,17,18]. Melanin synthesis is a conserved physiological process in insects that occurs in epidermal cells [19,20], and in the melanin synthesis pathway, the precursor DOPA (3, 4-dihydroxyphenylalanine) and dopamine are synthesized from tyrosine hydroxylase (TH) and DOPA decarboxylase (DDC) enzymes, respectively [21]. Tyrosine hydroxylase and DOPA decarboxylase enzymes are abound in central nervous system and epidermal cells of insects. Dopamine is an important catecholamine neurotransmitter in invertebrates and vertebrates [18]. In insects, dopamine is the major contributor for melanin biosynthesis, while in mammals DOPA is dominated [17,18]. The biochemistry of tyrosine metabolism is abounding in insects because tyrosine is the initiatory material of melanin formation. Tyrosine hydroxylase is the first key enzyme in this pathway that can convert tyrosine to DOPA. Then, DOPA decarboxylase catalyzes DOPA to dopamine. DOPA and dopamine are crucial precursors of sclerotization and melanization in insects. There was research reported that sclerotization takes precedence over melanization in insects. Dopamine firstly uses to synthesis NBAD by NBAD synthetase (*ebony*) and NADA by N-acetyldopamine transferase (*aaNAT*) for sclerotization. The redundant NBAD can be hydrolyzed back to dopamine by NBAD hydrolase (*Tan*). Then, the dopamine is oxidized by phenoloxidase (*laccase 2*) to dopaminequinone and further converted to melanin [17]. Many aspects of this process also influence the ecology, development, genetics, and physiology of insects [19]. Due to its important role in insect physiology, it is imperative to understand the role this plays in all vital processes through a comprehensive investigation of the characteristics of melanin synthesis.

The availability of sophisticated molecular techniques has enabled the disentangling of complex melanin synthesis processes in *H. axyridis* [22]. As previously reported, Ando et al. [3] delimited the melanin synthesis process of *H. axyridis* revealing that it begins immediately after eclosion in regions complementary to the red-pigment regions. Furthermore, we have previously demonstrated that dopamine melanin is the primary contributor in the elytra in *H. axyridis* through silencing *DDC* in the third instar larvae and *DDC* can regulate *H. axyridis* fecundity [23]. Recently, a single GATA transcription factor gene, *Pannier*, was identified in genome studies and was revealed to promote melanin production and suppress carotenoid synthesis during elytral development in *H. axyridis* [2,3]. *Pannier* is believed to regulate a subset of enzyme genes in the melanin synthesis pathway, because this process and genes of the enzymes that catalyze various chemical reactions are relatively conservative in animals. [3]. However, the specific mechanism of how *Pannier* regulates pigmentation patterns and the gene regulatory network remains unclear. The objectives of this study were to determine: (1) the melanin component in overall body of *H. axyridis*; (2) establish the synthesis time of melanin in *H. axyridis*; and (3) identify the regulatory mechanism of transcription factor *Pannier* on melanin synthesis in *H. axyridis*.

## 2. Results

### 2.1. Developmental Stage Expression Patterns of HaTH

Tyrosine hydroxylase (TH) is the first key enzyme in melanin synthesis, which catalyzes tyrosine converting into DOPA. The relative transcript levels of *HaTH* in each developmental stage were analyzed by RT-qPCR. Transcripts of *HaTH* were observed at all test stages and the highest levels of *HaTH* transcripts were detected on the fifth day of pupal development (pharate adult) (Figure 1).

### 2.2. The Melanin Synthesis Pathway in *H. axyridis*

Our previous study examining melanin synthesis pathways suggested that dopamine melanin is the major melanin in the elytra through silencing *HaDDC* in third instar larvae of *H. axyridis* [23]. In this study, we continue to investigate how the melanin synthesis pathway contributes to the overall body in *H. axyridis*. Due to the crucial role of *TH* in the melanin synthesis pathway, we first silenced *HaTH* in fourth instar larvae. The transcript levels of *HaTH* were significantly suppressed on days 2, 4, 6, and 8 after injection (Figure 2a). However, we did not find significant differences of pupation rate between *HaTH* silenced ladybirds and its control (Figure 2c). *HaTH* silenced pupae showed obvious abnormal phenotypes with a loss of melanic color throughout the body compared to respective controls (Figure 3). Furthermore, these pale pupae eclosed into the adult stage, but only 20% emerged into normal morphological adults (Figure 2d). This led to approximately 80% of abnormal eclosion, which was manifested with a deformed elytron with the puparium still attached in the body (Figure 4). In addition, melanin in the three subspecies (conspicua, sepectabilis, and succinea) of *TH* silenced ladybirds totally disappeared compared with their respective controls (Figure 3). We did not observe any movement or feeding in *TH* silenced ladybirds and they all died within 2 days of eclosion (Figure 2b).

DOPA decarboxylase (DDC) is the second important inverting enzyme that was investigated in our study. When fourth instar larvae of *H. axyridis* were injected with ds*HaDDC* at 300 ng /larva, the transcript levels of *HaDDC* were significantly suppressed at 2, 4, 6, and 8 days after injection (Figure 5a). However, we did not find significant differences of mortality, pupation rate, and eclosion rate between the *HaDDC* silenced group and their respective controls (Figure 5b–d). When *HaDDC* was silenced in fourth instar larvae, all injected ladybirds successfully pupated but melanin was not synthesized until 6h after pupation, later than their respective controls (Figure 3). Furthermore, although these pupae emerged into adults, melanin in the three subspecies (conspicua, sepectabilis, and succinea) of *H. axyridis* partially disappeared. Although the head and pronotum were normal, the elytral melanin was significantly decreased compared to respective controls (Figure 3).

### 2.3. The Critical Period of Melanin Synthesis in *H. axyridis*

In holometabolous insects, the pupal stage is considered the key material accumulation stage of melanization. To further delimit the critical period of melanin synthesis in *H. axyridis*, *TH* and *DDC* were silenced in 1-, 3-, and 5-day pupae. When the pupae (1-, 3-, and 5-day) were injected with ds*HaTH* (or ds*HaDDC*) at 300 ng /pupa, the transcript levels of *HaTH* (or *HaDDC*) were significantly suppressed on days 2, 4, and 6 compared to respective controls (Figure 6a–f). For the three subspecies of *H. axyridis*, 1- and 3-day pupal RNAi targeting *TH* resulted in complete disappearance of melanin in the overall body. However, when *TH* was silenced in 5-day pupae, the head and pronotum showed normal melanization with the control, but melanin in the elytra was significantly decreased (Figure 7). We obtained unexpected results with all three subspecies of *H. axyridis* showing normal melanization in the overall body even though transcript levels of *HaDDC* were significantly decreased after ds*HaDDC* injection (Figure 7).

### 2.4. The Regulatory Mechanism of Transcription Factor Pannier on Melanin Synthesis in *H. axyridis*

In a recent study, the transcript factor gene *Pannier* was identified as regulating melanin synthesis during the ontogeny process in *H. axyridis* [2,3]. To investigate the regulatory mechanism of *Pannier* on melanin synthesis, we first examined the function of *Pannier* on development in *H. axyridis* using RNAi. When fourth instar larvae of *H. axyridis* were injected with ds*HaPnr* at 300 ng /larva, transcript levels of *HaPnr* were significantly suppressed on days 2, 4, 6, and 8 after injection (Figure 8a). The fourth instar larvae of *HaPnr* that were suppressed successfully pupated, but the pupae showed an abnormal phenotype and melanin was absent from half of the body (Figure 8b). Furthermore, these pupae successfully eclosed into adults, but melanin totally disappeared from the overall body compared to controls (Figure 8b).

*TH* and *DDC* worked as important inverting enzymes, playing essential roles in the dopamine melanin synthesis process. To investigate the regulatory mechanism of *Pannier* on dopamine melanin synthesis in *H. axyridis*, we examined expression level of *TH* and *DDC* after *Pannier* was suppressed using RNAi. As mentioned above, results demonstrated that transcript levels of *HaPnr* were significantly suppressed on days 2, 4, and 6 after fourth instar larvae were injected with ds*HaPnr* (Figure 8a). These *HaPnr* suppressed samples were then used to detect expression levels of *TH* and *DDC*, and we revealed that expression levels of both *TH* and *DDC* were significantly decreased on day 8 after ds*HaPnr* injection (Figure 9a,b).

## 3. Discussion

The mechanism of melanin biosynthesis is conserved and has been well characterized in many insects [24,25]. The Asian multi-colored ladybird beetle, *H. axyridis*, exhibits intraspecific polymorphism based on melanization and is considered as an ideal model insect to investigate population diversity [1]. As previously reported, we have documented that dopamine melanin plays a significant role in elytral melanization of *H. axyridis* [23]. Recently, two independent studies reported that a GATA transcription factor gene, *Pannier*, regulates the highly diverse elytral color patterns in *H. axyridis* [2,3]. Based on these results, we performed gene-silencing analysis of the tyrosine metabolism pathway to investigate the molecular and regulatory mechanism of melanin synthesis in *H. axyridis*.

### 3.1. TH Has a Pleiotropic Role in the Development of *H. axyridis*

The first step in common melanin synthesis pathways is the hydroxylation of tyrosine to produce DOPA, and tyrosine hydroxylation (TH) catalyzes this reaction [17]. Firstly, developmental expression profiles of *HaTH* were analyzed by RT-qPCR and we revealed that transcripts of *HaTH* were dramatically increased in 5-day pupae (Figure 1). Several studies have demonstrated that *TH* is required in cuticle melanization, such as in *Drosophila melanogaster* (Diptera: Drosophilidae) [26], *Tribolium castaneum* (Coleoptera: Tenebrionidae) [27], *Papilio xuthus* (Lepidoptera: Papilionidae) [28], and *Bombyx mori* (Lepidoptera: Bombycidae) [20]. We obtained similar results with above studies, revealing that *HaTH* is also involved in melanin synthesis in *H. axyridis*. Suppression of the expression level of *HaTH* in fourth instar larvae and pupae (1-, 3-, and 5-day) resulted in the complete loss of melanic color in *H. axyridis* (Figure 3, Figure 7). In addition, our findings suggested that *TH* plays a predominant role in ecdysis and adult survival in *H. axyridis* because when *HaTH* expression was suppressed, abnormal eclosion occurred and this was lethal in the adult stage (Figure 2, Figure 4). Our results were consistent with previous reports that knockdown of tyrosine hydrolase enzymes drastically affects fundamental physiological processes such as embryogenesis, reproduction, ecdysis, and nymph survival in *Rhodnius prolixus* (Hemiptera: Reduviidae) [29] and *T. castaneum* [27]. The comprehensive analysis of this subject, coupled with our results and relative reports, confirms that *TH* is a pleiotropic gene in the development of insects.

### 3.2. Melanin in *H. axyridis* Is Synthetized from Dopamine

In all the insect species studied to date, melanin is primarily synthesized from dopamine, which differs from mammals where melanin is synthetized from DOPA [17,18]. In our previous study, through silencing *DDC* in third instar larvae, we demonstrated that dopamine melanin was the primary melanin in elytra during ontogenesis of *H. axyridis* [23]. To comprehensively investigate melanin synthesis in *H. axyridis*, *TH* and *DDC* were selected to be analyzed in fourth instar larvae. Our results showed that suppressed *HaTH* expression in fourth instar larvae resulted in complete disappearance of melanin in the overall body of pupae and three adult subspecies (Figure 3). These findings suggest that the melanin synthesis based on tyrosine metabolism plays a particularly significant role in the melanization of *H. axyridis*. Furthermore, when *HaDDC* was suppressed in fourth instar larvae, melanin was almost entirely absent in pupae and three adult subspecies, with the exception of their head and pronotum (Figure 3). Our results corroborate previous reports on *T. castaneum* [24], *Oncopeltus fasciatus* (Hemiptera: Lygaeidae) [30], *Periplaneta americana* (Blattodea: Blattidae) [31], and *R. prolixus* [29] that suppressed *DDC* expression resulting in loss of melanic color in most parts of the body. As described in the conserved melanin synthesis pathway, *DDC* converts DOPA to dopamine, hence, the enzyme is induced with suppressing and resulted in the irreversible loss of dopamine. Consequently, our results further confirm that dopamine melanin was the primary contributor to melanin synthesis in *H. axyridis*.

### 3.3. The Crucial Synthesis time of Dopamine Melanin Precursor in *H. axyridis*

*H. axyridis* is a holometabolous insect, having an identical appearance during the larval and pupal stages. The pupa is an important developmental stage in holometabolous insects because it is involved in the dissolving of larval tissue and adult organ reproduction. Ando et al. [3] reported that during elytral development process, the elytral primordia growth occurs in fourth instar larvae and detaches from the pupal exoskeletion and differentiation into adult elytra in the early pupal stage (24h after pupation) [22]. As demonstrated in our results, when *TH* and *DDC* were silenced in fourth instar larvae of *H. axyridis*, an obvious loss in melanic color of adults followed (Figure 3). To confirm if the same scenario is occurring in the pupal stage, further research is required to investigate the characteristic of melanin synthesis through the silencing of *TH* and *DDC* in 1-, 3-, and 5-day pupae, respectively. We found silencing *TH* expression in pupal stages resulted in complete loss of melanic color in the adult stage but silencing *DDC* resulted in normal adults compared to the control (Figure 7). These unexpected results were consistent with previous reports in *T. castaneum* that 1- to 2-day old pupae treated with ds*TcDDC* developed normally and molted successfully into the adult stage.Furthermore, injection of ds*TcDDC* in 1- to 2-day pupae did not result in any decrease in levels of dopamine [24]. We can therefore speculate the reasons for eclosion of normal adults, even though the remaining *HaDDC* transcript level was only 1.71% of the control (Figure 6d–f). Firstly, the RNAi technique has limited ability in mRNA expression regulation resulting in target gene residues. In addition, the highest levels of *HaDDC* transcripts were detected in 1-day pupae of *H. axyridis* [23]. From these two possible reasons, we can conclude that trace amounts of *HaDDC* transcript may remain after ds*HaDDC* injection in 1-day pupae and these residual *HaDDC* could complete the conversion of DOPA to dopamine. Other studies have demonstrated that melanin pigments are synthesized from non-pigmentary precursors and then expressed in the corresponding region [20]. In the melanin synthesis process, DOPA and dopamine serve as a precursor stored in epidermal cells, ultimately resulting in the melanic body coloration in insects [21]. Thus, we have reasons to believe that surplus *HaDDC* could complete the conservation of DOPA to dopamine in 1-day pupae (early pupae stage). Subsequently, dopamine served as a precursor stored in dorsal thin epidermis cells and then oxidases to quinone when melanization commences under the effect of phenol oxidase (*Laccase* 2).

### 3.4. The Regulation Mechanism of Pannier on Melanin Synthesis in *H. axyridis*

A review of melanin in *H. axyridis* concluded that the transcript factor gene *Pannier* can regulate melanin synthesis during the ontogeny process [22]. The function of *Pannier* promotes melanin synthesis in the expressed region and suppresses carotenoids in the ventral epidermal cell [2,3]. In the current study, we confirmed that dopamine melanin is the primary contributor to overall body melanization in *H. axyridis*. It is therefore important to further investigate the regulatory mechanism of *Pannier* on dopamine melanin synthesis in *H. axyridis*. Firstly, we investigated the function of *Pannier* on melanin synthesis in *H. axyridis* and documented that silencing *HaPnr* in fourth instar larvae resulted in partial loss of melanic color in pupae and complete loss of melanic color in adults (Figure 8b). Our results were consistent with previous research stating that *Pannier* is necessary to produce melanic color in *H. axyridis* [2,3]. Subsequently, expression levels of *TH* and *DDC* were detected after *HaPnr* was suppressed in fourth instar larvae. Our results showed that the expression levels of both *TH* and *DDC* were significantly decreased on day 8 after ds*HaPnr* injection (Figure 9a,b). These results indicated that suppressing the transcript levels of *Pannier* can significantly decrease expression levels of *TH* and *DDC*. These results also support our conclusion that *Pannier* could regulate the pivotal enzymes *TH* and *DDC* in the dopamine melanin synthesis pathway. Importantly, the regulatory mechanism of transcription factors on physiological pathways in further complex studies are therefore necessary to characterize whether other genes participate in the regulatory process.

## 4. Materials and Methods

### 4.1. Insects

*H. axyridis* were collected from cotton fields (GPS location: 39°95′ N, 116°28′ E) at experimental fields of the Beijing Academy of Agriculture and Forestry Sciences (BAAFS), Beijing, China. Insects were reared under standardized conditions following those described by Chen et al. [23]. Three subspecies (*conspicua*, *sepectabolis*, and *succinea*) were common at the field site.

### 4.2. Total RNA Isolation and cDNA Synthesis

Total RNA was isolated from the insects by using TRIzol reagent (Invitrogen, Carlsbad, CA, USA) and treated with “PrimeScript^TM^ RT reagent Kit with gDNA Eraser” (Takara, Dalian, China) to remove genomic DNA and synthesize the first-strand cDNA in a 20 μL reaction system according to the manufacturer’s instructions. The cDNA from untreated samples was used for the synthesis of double-stranded RNA (dsRNA) (*dsHaTH*, ds*HaDDC*, *dsHaPnr*) with specific primers.

### 4.3. Developmental Stage Expression profiles of HaTH

Developmental-stage-dependent expression profiles of *HaTH* were analyzed in all stage of *H. axyridis* at 12 time points, including embryos (1-, 2-, and 3-day eggs), larvae (1, 2, 3, and 4 instar larvae) and pupae (1-,2-, 3-,4-, and 5-day). There are three replicates and for each, 30 eggs, 10 first instar larvae, 3 second instar larvae, and fourth instar larvae and pupae were amalgamated as a biological sample. The total RNA was isolated from the insects by using TRIzol reagent and treated with “PrimeScriptTM RT reagent Kit with gDNA Eraser” (Takara, Dalian, China) to remove genomic DNA and synthesize the first-strand cDNA. Reverse transcription quantitative PCR (RT-qPCR) was performed to analyze the relative transcript levels of *HaTH* using “SYBR Green with the Applied Biosystems^®^ Real-time PCR Instrument” (ABI Laboratories, Hercules, California, USA). The transcript levels of *HaTH* were expressed as normalized transcript abundance using the ribosomal protein S49, *Harp49* (Accession number: AB552923) as an internal reference gene. The relative *HaTH* transcript levels were calculated according to the 2^−△△Ct^ method [32].

### 4.4. Synthesis of Double-stranded RNA (dsRNA)

Specific primers for ds*HaTH*, ds*HaDDC*, and ds*HaPnr* were designed using E-RNAi (http://www.dkfz.de/signaling/e-rnai3/idseq.php) (Table 1). The green fluorescent protein gene (*GFP*) was used as a control for the RNAi experiments, which was amplified from plasmid (pMD18-T simple Vector) using T7 promoter primers. The first chain of cDNA was amplified by Premix Taq^TM^ (TaKaRa, Dalian, China) with specific primers. The PCR products were purified by QIAquick^®^ PCR Purification Kit (QIAGEN, Hilden, Gemany) and used as templates of double-stranded RNAs (dsRNAs), which were synthesized using “MEGAscript^®^ RNAi Kit” (Invitrogen, Carlsbad, California, USA) according to the manufacturer’s instructions.

### 4.5. RNAi Experiments

Fourth instar larvae and pupae from the first, third, and fifth day were injected in the abdomen using Nanoject II injector (Drummond Scientific, Broomall, PA, USA) with 300 ng/individual of ds*HaTH*. The groups of control insects were treated with same dose. Similarly, ds*HaDDC* was injected into fourth instar larvae and pupae from the first, third, and fifth day. However, ds*HaPnr* was only injected in fourth instar larvae to explore the regulatory mechanism of *HaPnr* on melanin synthesis. The insects injected were fed on *Aphis craccivora* Koch (Hemiptera: Aphididae) under standard conditions for visual monitoring of phenotypes and other analyses. The RNAi experiment was performed with three biological replicates (each with at least 40 larvae) for each control and treatment.

### 4.6. Analysis of Expression Profiles by Reverse Transcription Quantitative PCR

To analyze the transcription of *HaTH* after injection, total RNA of ds*HaTH* and ds*GFP* were isolated from whole insects after the second, fourth, sixth, and eighth day. Each time point was analyzed with three biological samples and each sample was run with three technical replications. Total RNA was isolated from the insects using TRIzol reagent. Similar experimental procedures were undertaken on the insects injected in ds*HaDDC*. *Harp49* was used as the reference gene for these two experiments and the synthesis of cDNA was undertaken as previously described. RT-qPCR was performed with three biological replicates, each with three technical replicates. The insects injected with ds*Hapnr* were sampled to test relative transcript levels of *Hapnr*, *HaDDC* and *HaTH*. 

### 4.7. Image Processing

The phenotype of all insects in this experiment were observed and recorded by Zeiss Microscope SteREO Discovery V20 (Carl Zeiss, Oberkochen, Germany) with identical magnification, exposure time and intensity of light. 

### 4.8. Statistical Analysis

The analysis of transcript levels of *HaTH*, *HaDDC* and *HaPnr* in RT-qPCR were expressed as a percentage of the level in controls by dividing the relative expression value (REV) in dsRNA-injected insects by REV in ds*GFP*-injected insects and multiplying by 100. Percent data from developmental stage analysis and data from the RNAi experiments were arcsine square root transformed prior to analysis via SPSS (v. 22, IBM Corp. Armonk, NY, USA) followed by One-Way ANOVA (*p* < 0.05) to separate means.

## 5. Conclusions

In conclusion, our study provides an important new insight into melanization of *H. axyridis* suggesting that dopamine serves as a melanin precursor and could synthesize in the early pupal stage. Furthermore, we also suggest that the transcription factor *Pannier* could regulate the expression of *TH* and *DDC* in dopamine melanin synthesis pathways to promote melanin synthesis in *H. axyridis*.

## Figures and Tables

**Figure 1 ijms-21-02088-f001:**
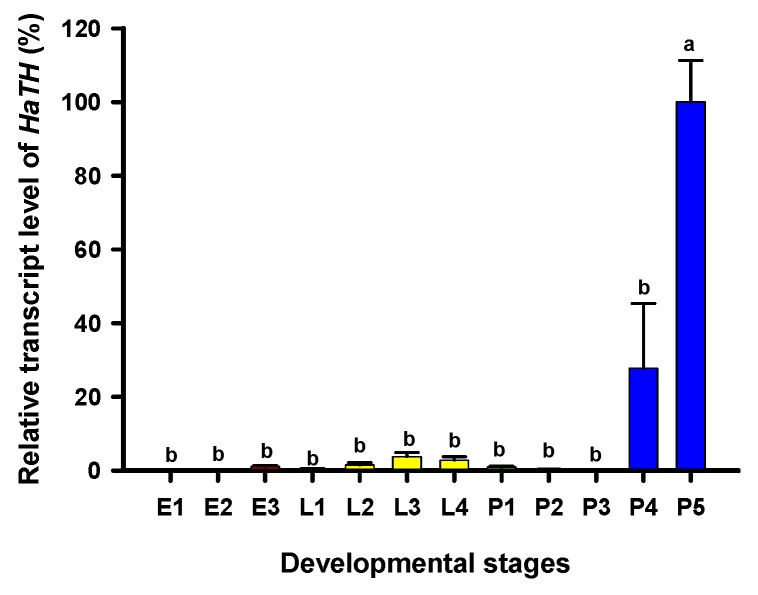
Relative transcript levels of *HaTH* at different developmental stages of *Harmonia axyridis* as determined by RT-qPCR. E1, E2, and E3 represent 1-, 2-, and 3-day eggs, respectively; L1, L2, L3, and L4 represent first, second, third, and fourth instar larvae, respectively; P1, P2, P 3, P4, and P5 represent 1-, 2-, 3-, 4-, and 5-day pupae, respectively. Different letters above the standard error bars indicate significant differences based on SPSS (v. 22, IBM Corp. Armonk, NY, USA) followed by One-Way ANOVA (*p* < 0.05). *H. axyridis* ribosomal protein 49 (*HaRp49*) was used as an internal reference gene to normalize the differences among the samples. Relative expression levels for *HaTH* were calculated based on the highest expressions of *HaTH* in 5-day pupae (P5) as 100% in the developmental stage.

**Figure 2 ijms-21-02088-f002:**
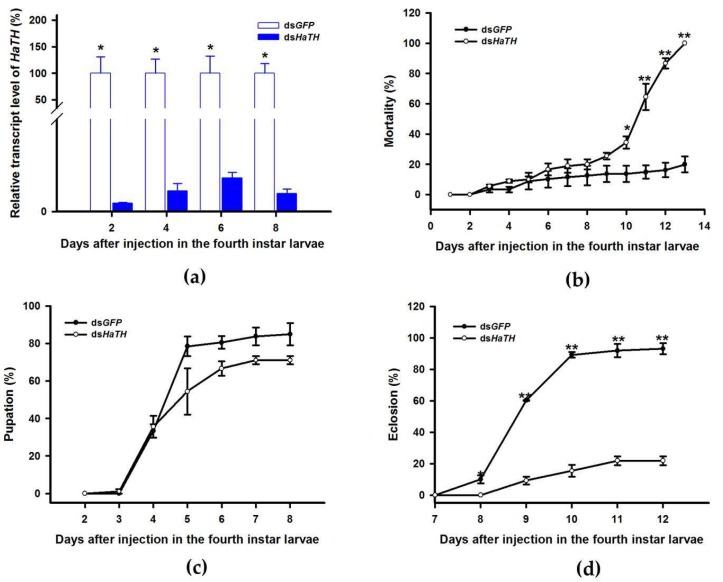
Time-dependent suppression of *HaTH* transcript in fourth instar larvae of *H. axyridis* injected with ds*HaTH* at 300 ng/larva or ds*GFP* at 300 ng/larva as determined by RT-qPCR (**a**), time-dependent mortality (**b**), pupation rate (**c**), and eclosion rate (**d**) in ds*HaTH* and ds*GFP*-injected larvae. The relative transcription levels (%) are presented as the mean and SE of three replicates with three insects. There were three replicates in the determination of mortality, pupation rate, and emergence rate, each of which had at least 30 fourth instar larvae. Notability analysis was based on SPSS (v. 22, IBM Corp. Armonk, NY, USA) followed by One-Way ANOVA (*p* < 0.05) within the same time point. Asterisk indicates a significant difference in the rate of eclosion and relative transcript level of *HaTH* between insect injected with ds*HaTH* and ds*GFP* (*: significant difference; **: extremely significant difference).

**Figure 3 ijms-21-02088-f003:**
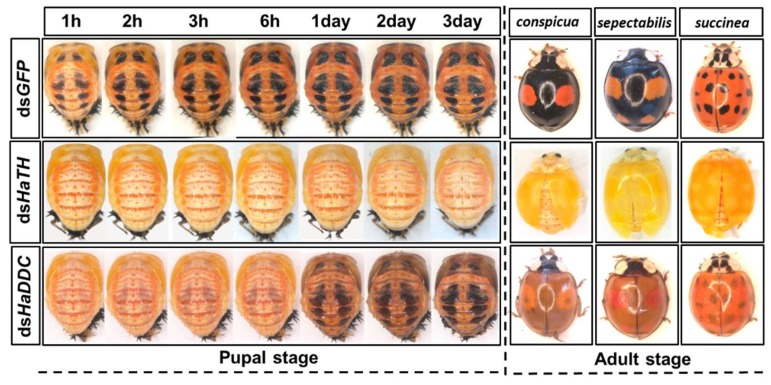
The phenotypes of abnormal pupae and adults from the fourth instar larvae of *H. axyridis* injected with ds*HaTH* and ds*HaDDC*, respectively.

**Figure 4 ijms-21-02088-f004:**
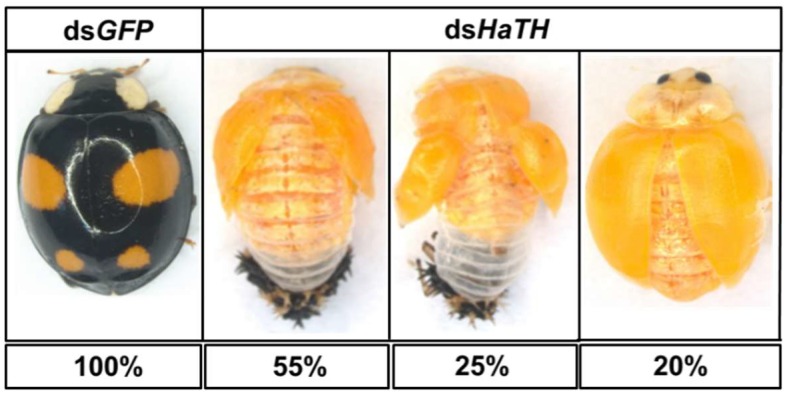
The phenotypes of an abnormal adult from the fourth instar larvae of *H. axyridis* injected with ds*HaTH*.

**Figure 5 ijms-21-02088-f005:**
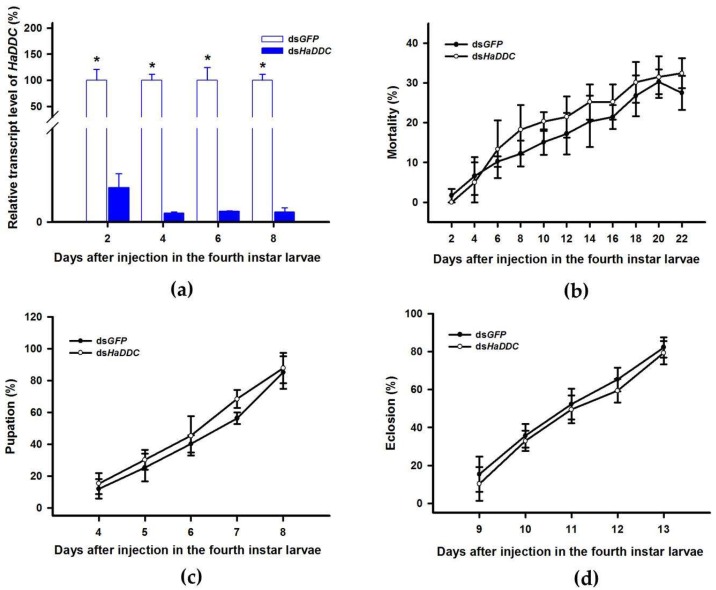
Time-dependent suppression of *HaDDC* transcript in fourth instar larvae of *H. axyridis* injected with ds*HaDDC* at 300 ng/larva or ds*GFP* at 300 ng/larva as determined by RT-qPCR (**a**), time-dependent mortality (**b**), pupation rate (**c**), and eclosion rate (**d**) in ds*HaTH* and ds*GFP*-injected larvae. The relative transcription levels (%) are presented as the mean and SE of three replicates with three insects. There were three replicates in the determination of mortality, pupation rate, and emergence rate, each of which had at least 30 fourth instar larvae. Notability analysis was based on SPSS (v. 22, IBM Corp. Armonk, NY, USA) followed by One-Way ANOVA (*p* < 0.05) within the same time point. “*” indicates a significant difference in relative transcript level of *HaDDC* between insect injected with ds*HaDDC* and ds*GFP*.

**Figure 6 ijms-21-02088-f006:**
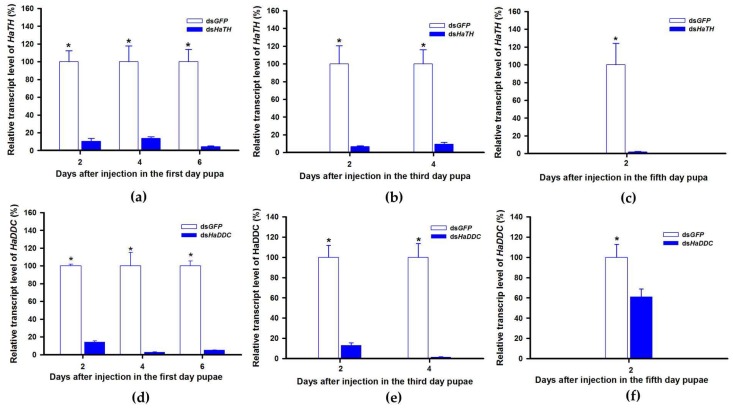
Time-dependent suppression of *HaTH* (**a**–**c**) and *HaDDC* (**d**–**f**) transcript in the 1-, 3-, and 5-day pupae of *H. axyridis* injected with ds*HaTH* and ds*HaDDC* at 300 ng/pupa or ds*GFP* at 300 ng/pupa as determined by RT-qPCR, respectively. The relative transcription levels (%) are presented as the mean and SE of three replicates with three insects. Notability analysis was based on SPSS (v. 22, IBM Corp. Armonk, NY, USA) followed by One-Way ANOVA (*p* < 0.05) within the same time point. “*” indicates a significant difference in relative transcript level of *HaTH* and *HaDDC* between insect injected with ds*HaTH* (or ds*HaDDC*) and ds*GFP*.

**Figure 7 ijms-21-02088-f007:**
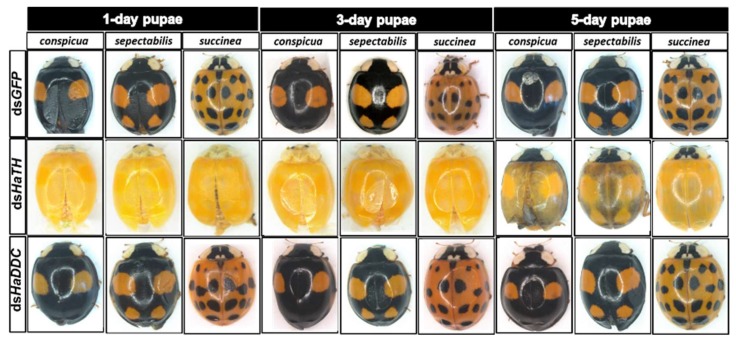
The phenotypes of adult from the pupae of *H. axyridis* injected with ds*HaTH* and ds*HaDDC*, respectively.

**Figure 8 ijms-21-02088-f008:**
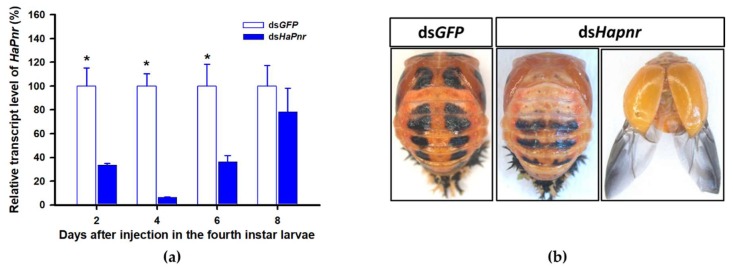
Time-dependent suppression of *HaPnr* transcript in the fourth instar larvae of *H. axyridis* injected with ds*HaPnr* at 300 ng/larva or ds*GFP* at 300 ng/larva as determined by RT-qPCR (**a**) and the phenotype of abnormal pupae and adult from fourth instar larvae injected with ds*HaPnr* (**b**). The relative transcription levels (%) are presented as the mean and SE of three replicates with three insects. Notability analysis was based on SPSS (v. 22, IBM Corp. Armonk, NY, USA) followed by One-Way ANOVA (*p* < 0.05) within the same time point. ”*” indicates a significant difference in relative transcript level of *HaPnr* between insect injected with ds*HaPnr* and ds*GFP*.

**Figure 9 ijms-21-02088-f009:**
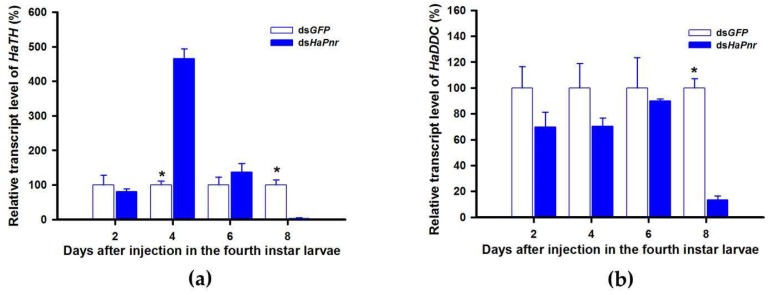
Relative transcript levels of *HaTH* (**a**) and *HaDDC* (**b**) following ds*HaPnr* injection in the fourth instar of *H. axyridis*. The results are presented as the mean and standard errors of three replicates; each was performed with a RNA sample prepared from three insects. Asterisk above the standard error bars indicate significant differences based on SPSS (v. 22, IBM Corp. Armonk, NY, USA) followed by One-Way ANOVA (*p* < 0.05) within the same time point. “*” indicates a significant difference in relative transcript level of *HaTH* and *HaDDC* between insect injected with ds*HaPnr* and ds*GFP*.

**Table 1 ijms-21-02088-t001:** Primers used to synthesize dsRNA and analyze transcript levels.

Application of Primers		Sequence (5’-3’)	Product Length (bp)
dsRNA synthesis	ds*GFP*(T_7_)-F	taatacgactcactatagggagaCAGTGCTTCAGCCGCTAC	288
	ds*GFP*(T_7_)-R	taatacgactcactatagggagaGTTCACCTTGATGCCGTTC	
	ds*Hapnr*(T_7_)-F	taatacgactcactatagggTACGAAAGTAACCCGTACCCC	398
	ds*Hapnr*(T_7_)-R	taatacgactcactatagggGGATGGTTTTGAGATGACGAA	
	ds*HaTH*(T_7_)-F	taatacgactcactatagggTATCCGGCAAGAAGACGTTC	479
	ds*HaTH*(T_7_)-R	taatacgactcactatagggGTGCAATGCTCCAGAACAGA	
	ds*HaDDC*(T_7_)-F	taatacgactcactatagggTATAAGGGAGAGGCGGGTTT	385
	ds*HaDDC*(T_7_)-R	taatacgactcactatagggGTAGCTTCACTCGCAGTCCC	
RT-qPCR	*Hapnr (Q)-F*	TACCCAGACCTTGGAACGAC	135
	*Hapnr (Q)-F*	CCGAAGATTTGCTGGTAAGG	
	*HaTH*(Q)-F	GCATCTTTGCTCCTGACA	83
	*HaTH*(Q)-R	AGGTCTAAGGGTGAATCCA	
	*HaDDC*(Q)-F	TAGTTGCCTTGCTTGGAG	187
	*HaDDC*(Q)-R	TTTGATTCGTCTGTGGGTA	
	*Harp49-F*	ACGGACTTCGGTAGGACG	**130**
	*Harp49-R*	CGCAGACAATCCCGAAA	

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
