# Peer review of "Molecular and Potential Regulatory Mechanisms of Melanin Synthesis in Harmonia axyridis"

_ijms, 2020, doi:10.3390/ijms21062088_

Round 1

Reviewer 1 Report

This study is an attempt to address the importance of melanization in insects.  The authors performed a gene-silencing analysis to understand the role of tyrosine hydroxylase and DOPA decarboxylase in the tyrosine metabolism pathway and melanin synthesis. The authors demonstrated that dopamine melanin was the primary contributor to the overall body melanization of H. axyridis. Their data suggest that dopamine serves as a melanin precursor for synthesis in the early pupal stage.

The subject of the manuscript is very narrow and the current version of the manuscript looks very specific. At the same time, the authors are trying to answer general questions about melanization. In my opinion, authors should revise the manuscript Introduction with a purpose to provide more detailed information about the biochemistry tyrosine metabolism pathway, the chemistry of melanin formation, and the roles of tyrosine hydroxylase and DOPA decarboxylase. This could make a paper available for a wider readership.

Author Response

We totally agree with the reviewer’s comment and have now added more information about the biochemistry tyrosine metabolism pathway, the chemistry of melanin formation, and the roles of tyrosine hydroxylase and DOPA decarboxylase in introduction part in our revised manuscript (Line 51- 62). In addition, we have revised our English expression with the help of a native English speaker.

Reviewer 2 Report

The manuscript is a very good work presenting interesting results. Manuscript title describes the article appropriately and lenght of the manuscript is adequate to all content. Description of materials and methods is clear, also results and discussion are coherent. The tables are clear and readable. The references are chosen correctly.

One thing needs correction: 1) in abstract line 19 mechanism instead of mechaniasm 2) line 45 product instead of production

Author Response

We are sorry for the mistakes:

1) We have changed the words “mechaniasm” to “mechanism” at Line 19 in our revised manuscript.

2) We have changed the words “production” to “product” in line 46 in our revised manuscript.

Round 2

Reviewer 1 Report

No other comments.